# Multi-Modality Sheep Face Recognition Based on Deep Learning

**DOI:** 10.3390/ani15081111

**Published:** 2025-04-11

**Authors:** Sheng Liao, Yan Shu, Fang Tian, Yong Zhou, Guoliang Li, Cheng Zhang, Chao Yao, Zike Wang, Longjie Che

**Affiliations:** 1College of Informatics, Huazhong Agricultural University, Wuhan 430070, China; stls@webmail.hzau.edu.cn (S.L.); 158008825@webmail.hzau.edu.cn (Y.S.); fangzhihai_2003@mail.hzau.edu.cn (F.T.); guoliang.li@mail.hzau.edu.cn (G.L.); 2Key Laboratory of Smart Farming for Agricultural Animals, Huazhong Agricultural University, Wuhan 430070, China; 3Hubei Engineering Technology Research Center of Agricultural Big Data, Huazhong Agricultural University, Wuhan 430070, China; 4Engineering Research Center of Intelligent Technology for Agriculture, Ministry of Education, Wuhan 430070, China; 5Jinchang City Animal Husbandry and Veterinary Station of Gansu Province, Jinchang 737101, China; jcnmzhouy@163.com; 6School of Information, Wuhan Vocational College of Software and Engineering (Wuhan Open University), Wuhan 430205, China; 7Gansu Animal Husbandry Technology Extension General Station, Lanzhou 730030, China; gsxmzzwzk1977@163.com (Z.W.); chelongjie123@163.com (L.C.)

**Keywords:** sheep face recognition, multimodal, deep learning, ResNet, CBAM attention

## Abstract

Identifying sheep faces presents significant challenges due to their morphological similarities and the effects of varying lighting conditions and angles on image quality. This study introduces a novel model aimed at enhancing recognition accuracy by integrating color (RGB) and depth data. This model effectively learns from both the geometric features present in depth images and the texture features found in color images. Experimental results indicate that this model significantly improves recognition accuracy, even under complex lighting conditions and varying angles. This technology holds considerable promise for applications such as farm management and animal tracking, where precise identification is crucial.

## 1. Introduction

With the advancement of modern animal husbandry, the demand for enhanced breeding efficiency is increasing. Sheep are significant agricultural animals that play a crucial role in both mutton and wool production. However, traditional sheep management practices frequently depend on manual observation and intervention. For instance, sheep identification is typically achieved through the use of ear tags or other identification markers [1,2,3]. This method can induce stress and discomfort in the sheep while also elevating the labor intensity for the staff.

In recent years, numerous scholars have investigated the application of neural networks for the contactless identification of animals [4,5,6,7]. In the realm of sheep face recognition, Zhang et al. proposed a model based on an improved version of AlexNet [8]. Similarly, Wan et al. developed a sheep face recognition model utilizing deep learning and bilinear feature fusion [9]. Additionally, Zhang Hongming’s team introduced a model that integrates MobileFaceNet with efficient channel coalesce spatial information attention, achieving a recognition rate of 96.73% in closed set verification and 88.03% in open set verification [10]. Zhang Shilong et al. presented a sheep face recognition method that employs weak edge feature fusion [11]. This approach combines and classifies features extracted by two networks through training a weak edge feature fusion network alongside a backbone feature extraction network, resulting in a significant improvement in recognition accuracy compared to the original network lacking fused edge features. Furthermore, Ning Jifeng et al. proposed a method for the individual identification of dairy goats based on an improved YOLOv5s (You Only Look Once version 5 small) model [12]. This method first employs transfer learning to pre-train the YOLOv5s detection model, subsequently incorporating the SimAM attention module in the feature extraction layer and the CARAFE (Content-Aware ReAssembly of FEatures) upsampling module in the feature fusion layer to enhance the restoration of facial details. Experimental results indicate that the average accuracy of the improved YOLOv5s model is 97.41%, which represents an increase of 2.21 percentage points over the original YOLOv5s model. Li proposed the MobileViTFace model [13], which combines convolutional neural networks with Vision Transformer (ViT). In comparison to the standard ViT model, this approach requires less training data and exhibits lower computational complexity, facilitating easier deployment on edge devices. The model achieved a recognition accuracy of 97.13% on a dataset of 7434 sheep face images, encompassing 186 individual sheep. Zhang introduced an efficient multi-angle sheep face recognition method [14]. They developed a multi-view sheep face image collection device, utilizing 50 experimental sheep to create a multi-view sheep face dataset. Furthermore, they established a high-precision sheep face recognition model, T2T-ViT-SFR, by integrating various optimization strategies to enhance model performance.

Previous research has demonstrated that sheep face recognition methods based on deep learning facilitate simpler data collection and can achieve higher accuracy without causing harm to the animals. Additionally, these methods offer advantages such as low cost and ease of operation. As a crucial identification tool, sheep face recognition technology significantly enhances management efficiency and promotes the modernization of animal husbandry. However, this technology faces numerous challenges. Although traditional sheep face recognition methods utilize neural networks for non-contact recognition, they rely solely on RGB images to extract features. In practical applications, RGB images can be influenced by factors such as ambient light and angle, leading to limited anti-interference capabilities of the model. In the realm of face recognition, numerous studies have explored the integration of multi-modal data for improved recognition [15,16,17,18]. In particular, Kaashki et al. proposed a three-dimensional face recognition method designed for complex conditions, based on a three-dimensional constrained local model (CLM-Z) [19]. This method employs CLM-Z to model and detect key facial points, subsequently describing these points with oriented gradient histograms, local binary patterns, and 3D local binary patterns, before utilizing support vector machines for face recognition.

Uppal et al. proposed a two-level attention mechanism that integrates features from both RGB and depth modalities [20]. The first-level attention mechanism operates within each modality, while the second-level attention mechanism is employed during the fusion process, effectively combining the features of the two modalities. Chen et al. utilized a 3D deformation model to generate high-quality virtual depth data corresponding to RGB images and introduced an adaptive confidence weighting mechanism during the inference stage [21]. This mechanism dynamically adjusts each modality based on the feature extraction results from both RGB and depth modalities, thereby enhancing the confidence weight of the state. Ultimately, modal fusion through weighted similarity scores significantly enhances the performance of the RGB-D face recognition system, particularly when dealing with low-quality depth images. Grati et al. proposed a novel approach for learning local feature representations from two modalities [22]. Unlike traditional global feature extraction methods, this approach emphasizes the extraction of features from local areas and combines these features through a deep learning model, thereby capturing finer details and variations of the face, which improves recognition accuracy and robustness. Zhang et al. introduced a dual-branch face recognition method based on the InceptionV2 network, which learns complementary features from multiple modalities and designs a common feature space that maps different modalities to the same feature representation [23]. This transformation enables the model to achieve cross-modal matching capabilities. Currently, in the field of face recognition, the fusion of multi-modal data has effectively enhanced the performance of recognition models. However, there remains a relative scarcity of multi-modal recognition methods in the domain of sheep face recognition.

This paper proposes a dual-branch multi-modal sheep face recognition model that integrates depth and RGB modalities. The depth image provides the distance from each pixel of the sheep’s face to the camera, thereby capturing the geometric information of the sheep’s face, which is less susceptible to environmental interference. Simultaneously, the model combines data from both modalities, utilizing geometric information to enhance the texture information in the RGB image. This approach effectively improves the model’s resilience to interference and enhances recognition accuracy.

## 2. Materials and Methods

### 2.1. Data Acquisition and Processing

#### 2.1.1. Multi-Modal Data Acquisition

To enhance the robustness of the model, two distinct datasets were collected independently. The first dataset comprises Hu sheep data gathered from a sheep farm located in Huangpi, Wuhan, Hubei Province. The second dataset includes information on military reclamation white sheep and white Suffolk sheep, collected at the experimental sheep farm of the Xinjiang Academy of Agricultural and Reclamation Sciences in Shihezi City, Xinjiang Uygur Autonomous Region. These datasets were acquired under varying lighting conditions. Depth data and RGB data were simultaneously collected using the Microsoft Kinect V2 depth camera, which operates based on the time-of-flight (TOF) [24] ranging principle. This camera emits a laser and measures the time difference between light emission and its reflection from the sheep’s face back to the camera, thus determining the distance to the sheep’s face and generating a depth image. The original depth image is a 16-bit grayscale image with a resolution of 640 × 576 pixels, while the RGB image is an 8-bit color image with a resolution of 1920 × 1080 pixels. Consequently, it is essential to visualize the depth map and align it with the RGB image. First, the intrinsic and extrinsic matrices of both the RGB and depth cameras are obtained through calibration. Next, the depth map is projected into 3D space using its pixel coordinates and depth values, and transformed into the RGB camera coordinate system via the extrinsic matrix, as illustrated in Equation (Equation 1). Following this, the 3D points are projected back onto the 2D image plane using the intrinsic parameters of the RGB camera, as shown in Equation (Equation 2), to obtain the aligned pixel coordinates. Finally, nearest-neighbor interpolation is applied to resample the depth map to match the resolution of the RGB image, achieving pixel-level alignment.(1)$$P_{3D,RGB}=R\left(K_{d}^{-1}\left[\begin{array}{c} u_{d}\\ v_{d}\\ 1 \end{array}\right]d\left(u_{d},v_{d}\right)\right)+T$$(2)$$\left[\begin{array}{c} u\\ v \end{array}\right]=\frac{1}{\omega }K_{r}P_{3D,RGB}$$

Here, $P_{3D,RGB}$ represents the 3D point coordinates projected onto the RGB camera coordinate system. The extrinsic matrix of the RGB camera is denoted by *R* and *T*, where *R* is the rotation matrix and T is the translation vector, which are utilized to transform the 3D point cloud from the depth camera to the RGB camera coordinate system. The term $K_{d}^{-1}$ refers to the inverse matrix of the depth camera’s intrinsic parameters, which is employed to convert pixel coordinates into normalized camera coordinates. The coordinates $\left(u_{d},v_{d}\right)$ indicate the pixel locations in the depth image, while $d\left(u_{d},v_{d}\right)$ signifies the depth value corresponding to that pixel. The coordinates $\left(u,v\right)$ represent the pixel positions after aligning the depth map with the RGB image, ω is the normalization factor obtained from the projection, and $K_{r}$ is the intrinsic matrix of the RGB camera, which is used to project the 3D coordinates back onto the pixel plane of the RGB image. The aligned RGB image and the corresponding depth map, visualized in 8-bit format, are presented in Figure 1.

Due to the requirement of initializing the depth camera each time a picture is taken, along with the necessary interval between each capture, acquiring images sequentially would be time-consuming for data collection. Consequently, we employed a method of video recording followed by extraction of frames. The Kinect camera facilitates the recording of both depth and RGB dual-track video, with each sheep’s video lasting for 20 s at a rate of 30 frames per second. After recording, the RGB images and their corresponding depth images can be extracted from the two tracks, respectively.

#### 2.1.2. Dataset Construction

Initially, we utilized MKVToolNix software (version 80.0.0) to conduct preliminary editing on the dual-track video data, cropping out sections that did not feature the sheep’s face. Subsequently, we employed FFmpeg to extract 5 frames per second from the recorded video, converting these frames into images of the sheep’s face while discarding those that were too similar or of low quality. Ultimately, we obtained a total of 99 sheep, resulting in 12,504 sets of images; each set comprised RGB images alongside their corresponding depth images. From this collection, 500 images were randomly selected from various sheep data and annotated for training a YOLOv8n [25] detection model. An example of the model’s detection output is presented in Figure 2.

To extract the sheep face from the RGB data, we first utilize the trained YOLOv8n model to predict the position of the sheep face frame. Subsequently, we isolated all pixels outside this frame. This method allows us to extract only the pixels corresponding to the sheep’s face in the RGB data. Since the depth map aligns with the RGB image, the pixels from both images correspond one-to-one. Therefore, by extracting the depth map at the same coordinates, we can obtain data that exclusively contains the sheep face area. The processing flow is illustrated in Figure 3.

After processing, the depth map and the corresponding RGB image are presented in Figure 4, where the image on the right highlights the filtered depth area of the sheep’s face. The depth map is measured in millimeters.

Divide all data consisting solely of sheep faces into training and validation sets in an 80:20 ratio. The outcomes of this division are presented in Table 1.

### 2.2. Identification Methods

#### 2.2.1. Two-Stream Convolutional Network Structure

Two-stream convolutional networks [26] represent a prevalent and effective architectural design in neural networks, particularly in applications involving multi-modal data fusion, feature extraction, and task decomposition. This structure typically comprises two independent branches, each tasked with processing distinct types of inputs or different segments of features. The fundamental concept is to manage data from various modalities through separate pathways, thereby preserving the unique characteristics of each modality. In the model proposed in this article, a dual-branch structure is employed at the initial stage, as illustrated in Figure 5. Each branch independently processes RGB data and depth data, consisting of a 7×7 convolutional layer followed by two Inception V2 [27] convolutional layers. The architecture of Inception V2 is depicted in Figure 5. This configuration enables the simultaneous capture of spatial features from both modalities at varying scales while maintaining computational efficiency by segmenting the input feature map into multiple parallel convolution branches (including 1×1 and 3×3 convolutions) and pooling operations.

#### 2.2.2. ResNet18 Network

This paper presents enhancements based on the ResNet [28] network architecture. ResNet is a deep residual network developed to address the degradation issues that arise during the training of deep neural networks. The core structure of ResNet consists of multiple residual blocks (Basic Blocks), each containing two 3×3 convolutional layers, followed by a batch normalization layer and a ReLU activation function. Within each residual block, the input is directly added to the output by skip connection and is then activated by ReLU. This design preserves essential input information without imposing additional computational burdens, thereby alleviating the common problem of gradient vanishing in deep networks.

ResNet18 comprises four major layers, each consisting of two Basic Blocks, totaling 18 layers. The number of channels gradually increases between each layer, enabling the network to capture more detailed feature information.

In the multi-modal sheep face recognition model, ResNet18 is selected due to its hierarchical structure, which can capture sufficient feature information without incurring excessive computational complexity. Particularly in multimodal data processing, the Basic Blocks of ResNet18 can effectively fuse RGB and depth information, ensuring that critical information is not lost during feature extraction. Furthermore, the design of skip connections enhances the network’s robustness in managing complex data, thereby improving the model’s recognition accuracy and efficiency.

### 2.3. Attention Module

#### 2.3.1. Convolutional Block Attention Module

The Convolutional Block Attention Module (CBAM) [29] is a lightweight and versatile attention mechanism that is extensively utilized in various convolutional neural networks to enhance the network’s focus on important features. The CBAM module primarily consists of a channel attention module and a spatial attention module, which are connected in series. By performing attention calculations on input features across both spatial and channel dimensions, CBAM significantly enhances the feature expression capabilities of the model while maintaining lower computational costs. This module can be seamlessly integrated into existing network architectures in a modular fashion, thereby improving network performance without considerably increasing computational complexity. This article employs the CBAM module to fuse RGB and depth modalities due to the distinct characteristics of these two modalities. The channel attention of CBAM can amplify the network’s focus on specific features within the RGB and depth modalities, while spatial attention enhances the network’s ability to capture key information locations within the image. Initially, a channel attention weight ($W_{CA}$) is calculated, with the calculation formula presented in Equation (Equation 3).(3)$$W_{CA}=\sigma \left(MLP\left(AvgPool\left(X\right)+MLP\left(MaxPool\left(X\right)\right)\right)\right)$$

In this formula, *X* represents the concatenation of the feature maps produced by both the RGB branch and the depth branch. The AvgPool operation denotes global average pooling, while MaxPool indicates global maximum pooling. Additionally, MLP refers to a multi-layer perceptron, and σ signifies the Sigmoid activation function.

Channel attention weights ($X_{CA}$) are applied, as demonstrated in Equation (Equation 4).(4)$$X_{CA}=X\times W_{CA}$$

Subsequently, a spatial attention weight ($W_{SA}$) is computed using the formula presented in Equation (Equation 5).(5)$$W_{SA}=\sigma \left(Conv\left(Cat\left(AvgPool\left(X_{CA}\right),MaxPool\left(X_{CA}\right)\right)\right)\right)$$

In this formula, Conv denotes the convolution operation, while Cat signifies the concatenation of the results from average pooling and max pooling along the channel dimension.

Finally, by applying the spatial attention weights, as illustrated in Equation (Equation 6), we obtain the fused feature map of the two modalities.(6)$$X_{SA}=X\times W_{SA}$$

#### 2.3.2. Mamba Module

Mamba [30] is a linear-time sequence modeling approach that employs selective state spaces. This approach establishes connections between global and local features, thereby enhancing the model’s expressiveness in processing multi-modal data. Given the potential emergence of irrelevant or redundant features following multi-modal data fusion, Mamba effectively prioritizes significant feature information while suppressing the influence of less important features through its selective mechanism. This article integrates the Mamba module into the ResNet Basic Block, facilitating the processing of the fused data. The improved basic block structure is illustrated in Figure 6, demonstrating its effectiveness in enhancing the model’s robustness and mitigating interference from non-sheep face components.

#### 2.3.3. A Multimodal Sheep Face Recognition Network

Due to the susceptibility of single RGB modal data to environmental and angular interference, the integration of RGB and depth modalities can significantly enhance the robustness and accuracy of sheep face recognition. However, traditional neural networks are unable to process multi-modal data concurrently. To address this limitation, we propose a dual-branch structure based on ResNet18, resulting in a multi-modal sheep face recognition network that incorporates the CBAM attention module and the Mamba module. This model is referred to as CBAM-DualRESNetMamba, or CBAM-DRESM for short. The specific structure of the model is illustrated in Figure 7.

Initially, a dual-branch structure is employed in the early stages of the network, enabling the model to process data from two modalities independently. This configuration ensures that the network parameters associated with each modality do not interfere with one another, thereby allowing for the distinct characteristics of both modalities to be learned effectively. Additionally, we incorporate two Inception V2 layers following the 7×7 large convolution kernel of the original ResNet18 architecture, while large convolution kernels can swiftly reduce the size of feature maps, they often fail to capture the subtle features necessary for distinguishing different sheep. The CBAM-DRESM structure was designed to capture the characteristics of sheep face images across various modalities and scales. This not only compensates for the detail-capturing limitations of large convolution kernels but also facilitates effective fusion through multiple parallel branches in Inception V2, integrating both local and global features.

Subsequently, the CBAM module is integrated in the middle stages of the network to merge the two modalities. By leveraging both channel attention and spatial attention, the CBAM module enhances the network’s focus on critical channels and spatial regions during the fusion of different modalities. This approach allows for improved acquisition of complementary information, ensuring that the two modalities can integrate effectively and complementarily. In the later stages of the network, the Mamba module is incorporated into the ResNet basic block to bolster the network’s ability to learn complementary features post-fusion. The Mamba module serves as an advanced attention mechanism that enhances the feature selectivity of the network, enabling a more effective focus on features pertinent to the target task.

Finally, four loss functions are constructed to evaluate the model’s performance and accuracy. (1) Two independent loss functions, loss_rgb and loss_depth, are established for the RGB and depth modalities, respectively. These loss functions are employed to optimize the feature extraction capabilities of each modal branch, ensuring that each modality operates independently and effectively. (2) A third loss function, loss_X, is derived from the fused feature vector. This loss function assesses and optimizes the complementary fusion effect between the two modalities, ensuring that key information is retained and enhanced during the fusion process. (3) The outputs of the three fully connected layers are combined to create a comprehensive loss function, loss_fusion. The four loss functions are weighted and summed to yield the total loss function, which serves to globally assess the overall performance of the model. The utilization of multiple loss functions facilitates backpropagation, thereby better guiding the optimization of parameters and enhancing recognition accuracy and robustness during model training.

## 3. Experiment and Results Analysis

### 3.1. Experimental Setup and Evaluation Metrics

#### 3.1.1. Experimental Parameter Setting and Experimental Process

Experimental equipment configuration: The system operates on an X86_64 Linux platform, utilizing an NVIDIA TITAN Xp graphics card and CUDA version 11.8. The software environment comprises Python 3.9, with the Pytorch deep learning framework employed for model construction, specifically version 1.10.1. The model training parameters include the use of the SGD (Stochastic Gradient Descent) optimizer and the AAM-Softmax (Angular Additive Margin Softmax) loss function. The initial learning rate is set at 0.01, with the cosine annealing algorithm implemented to dynamically adjust the learning rate. The batch size is established at 128, and the epochs are set to 100.

Experimental process: Following the editing of the dual-track video, RGB and depth images were extracted. Utilizing the SDK provided with the Kinect camera, and leveraging the camera’s internal and external parameters, we aligned the depth and RGB images to obtain a fully aligned pixel set. Subsequently, 500 images were randomly selected for annotation, and the YOLOv8n sheep face detection model was trained. This model achieved a detection accuracy of 99.9% and effectively extracts the sheep face region. Finally, all sheep face images were partitioned into training and validation sets in an 8:2 ratio, ensuring a one-to-one correspondence between the depth map and the RGB images.

#### 3.1.2. Evaluation Indicators

In sheep face recognition experiments, Accuracy, F1-Score, and FRR (False Reject Rate) are commonly employed as evaluation indicators for the recognition model. Accuracy is the most straightforward and transparent indicator, with its calculation formula presented in Equation (Equation 7).(7)$$Accuracy=\frac{\sum_{i=1}^{k}TP_{i}}{\sum_{i=1}^{k}\left(TP_{i}+FP_{i}+FN_{i}\right)}$$

In this formula, $TP_{i}$ denotes the number of samples that the model correctly predicts as belonging to the class *i*, $FP_{i}$ indicates the number of samples that the model incorrectly predicts as the i-th class, and $FN_{i}$ represents the number of samples that the model erroneously classifies as non-i-th class.

F1-Score is an important metric in machine learning and statistics, used to assess the performance of classification models. It combines precision and recall, representing their harmonic mean. The formula for Precision is provided in Equation (Equation 6), the formula for Recall is shown in Equation (Equation 9), and the formula for F1-Score is outlined in Equation (Equation 10).(8)$$Precision=\frac{TP}{\left(TP+FP\right)}$$(9)$$Recall=\frac{TP}{\left(TP+FN\right)}$$(10)$$F1-Score=2\times \frac{Precision\times Recall}{\left(Precision+Recall\right)}$$

Both Precision and Recall in Equations (Equation 8) and (Equation 9) utilize the macro-average calculation method, whereby the Precision for each category is computed and subsequently averaged across all categories.

FRR quantifies the likelihood that the model erroneously rejects samples that genuinely belong to known categories, categorizing them as unknown. This metric is crucial for evaluating the model’s usability. The formula for the rejection rate is illustrated in Equation (Equation 11).(11)$$FRR=\frac{FN}{\left(TP+FN\right)}$$

### 3.2. Performance Comparison of Different Models

In recent years, the accuracy comparison between commonly used models in the field of deep learning and the CBAM-DRESM model constructed in this paper is presented in Table 2. This model aims to fuse RGB and depth modalities for sheep face recognition; thus, the accuracy of other models, when inputting only a single modality, was compared. The images of the two modalities were processed through two model branches, followed by an accuracy assessment of the splicing recognition. As shown in Table 2, the CBAM-DRESM model proposed in this article outperforms the baseline model ResNet18 [28], as well as the control group models MobieNetV2 [31], VGG-16 [32], EfficientNetV2-S [33], Vision Transformer(ViT)-B [34], and ConvNeXt-T [35]. Specifically, the accuracy of this model increased by 2.17% compared to the original ResNet-18 which only inputed a single RGB modality and by more than 2.64% compared to the other models. Furthermore, Table 2 indicates that the accuracy of most models when inputting two modalities is lower than that of a single RGB modality. This phenomenon can be attributed to the fact that simple vector splicing does not account for the complex interrelationships between multiple modalities, treating both modalities equally. Consequently, each modality fails to leverage its unique characteristics, leading to ineffective integration of the features from the two modalities. In contrast, CBAM-DRESM independently processes the two modalities through a dual-branch structure and incorporates an attention mechanism to adaptively adjust the weights of the two for effective fusion.

The comparison of F1-Scores across different models is illustrated in Figure 8, which also evaluates the three input types. Figure 8 demonstrates that the CBAM-DRESM model achieves the highest F1-Score, indicating superior performance.

With the recognition threshold set at 0.8, the false rejection rate (FRR) of various models is illustrated in Figure 9. Evidently, the FRR of the CBAM-DRESM model is notably lower than that of other models, suggesting that the CBAM-DRESM model also exhibits superior usability.

### 3.3. Ablation Experiment

To verify the impact of the two-stream convolutional network structure, attention module, and multi-loss function design on the recognition accuracy of the CBAM-DRESM model constructed in this study, we conducted an ablation experiment. The test involved gradually adding model components, with the experimental results presented in Table 3. Initially, a two-stream convolutional network structure was integrated into the original ResNet18, and multiple loss functions were employed to train and evaluate the model. The results indicated an approximate increase of 0.4% in both the model’s identification accuracy and F1-Score. This finding suggests that the two-stream convolutional network structure effectively processes two modalities independently, thereby enhancing the model’s capacity to extract multi-modal data features. Subsequently, following the introduction of the CBAM attention module, a significant reduction in the FRR was observed. Data comparison reveals that this module enables the successful identification of sheep that were previously classified as unknown in complex scenarios. This indicates that the integration of the CBAM module facilitates a better combination of the characteristics from the two modalities. Finally, after incorporating the Mamba module, the identification accuracy of the complete model, which includes all modules, reached a peak of 98.49%, while the FRR decreased to its lowest point of 1.5%. This demonstrates that the Mamba module enhances the learning of critical aspects of complementary features.

### 3.4. Comparative Experiments on Attention Mechanisms

To validate the performance of the attention module in this experiment, we compared the effects of various attention mechanisms on recognition accuracy during both the feature fusion stage and the complementary feature enhancement learning stage. The evaluated attention mechanisms include Squeeze-and-Excitation (SE) [36], Efficient Channel Attention (ECA) [37], Self-Attention (SA) [38], and Coordinate Attention (CA) [39], all of which were employed at identical embedding positions. The experimental results are summarized in Table 4 and Table 5. During the feature fusion stage, the CBAM model achieved the highest performance, with a recognition accuracy of 98.13% and a rejection rate of 1.89%. Its channel-spatial joint attention mechanism significantly enhanced the capability for multimodal feature fusion. CA and ECA ranked second and third, respectively, while SA, due to its global modeling approach that diminishes local discriminative features, exhibited the lowest performance. In the complementary feature enhancement learning stage, the introduction of the selective state space model Mamba resulted in the highest accuracy of 98.49%, demonstrating that Mamba possesses superior learning capability in selecting the relevance of complementary features. Consequently, this paper selects the CBAM+Mamba attention combination to construct the multimodal sheep face recognition model.

## 4. Discussion

Although current research on sheep face recognition achieved high recognition accuracy through improvements in network architectures and feature fusion strategies, existing methods predominantly rely on a single RGB modality. This reliance poses significant challenges in complex agricultural scenarios due to the modality’s limited anti-interference capabilities, while multimodal fusion techniques have shown advantages in illumination robustness and geometric perception within the realm of face recognition, their potential application in agricultural biometrics remains largely underexplored. This study introduces an RGB-D multimodal framework, which for the first time validates the complementary value of depth data in biometric scenarios, achieving a high recognition accuracy of 98.49%. This result not only demonstrates the feasibility of multimodal fusion in agricultural biometrics but also provides an extensible technical pathway for future research in this field.

However, during the experimental process, several issues were identified that require further research and improvement, as detailed below.

### 4.1. Optimal Shooting Distance for 3D Cameras

Prior to the experiment, we observed that the depth maps of human faces captured by the Microsoft Kinect V2 camera (Seattle, DC, USA) were more complete and detailed when the target was positioned within a distance range of 0.8 to 2 m from the camera. Consequently, we adopted this distance range as the shooting distance for subsequent research. However, this conclusion has yet to be systematically evaluated. In future work, we will conduct more comprehensive experiments to thoroughly investigate the optimal shooting distance for 3D cameras in sheep face recognition, aiming to provide more precise guidance for practical applications.

### 4.2. Selection and Compatibility Issues of 3D Cameras

Given that the Microsoft Kinect V2 camera has been discontinued, our team has tested various types of 3D cameras to ensure the sustainability of the research. The cameras evaluated include Hikrobot MV-DT01SDU ToF camera (Hangzhou, China), Intel^@^ RealSense^TM^ Depth Camera D435f (Santa Clara, CA, USA), and Orbbec Femto Bolt 3D camera (Shenzhen, China). After comprehensive consideration of factors such as performance and cost-effectiveness, we ultimately selected Orbbec Femto Bolt 3D camera as a replacement for the Microsoft Kinect V2 camera. This camera not only offers excellent performance but is also compatible with the underlying architecture of the Azure Kinect SDK, effectively reducing the technical barriers for system migration and providing strong support for the smooth progression of the research.

### 4.3. Diversity of Sheep Breeds

The sheep breeds primarily collected in this study are Hu sheep, Junken White sheep, and White Suffolk sheep, all of which predominantly have white coats. Systematic testing has not yet been conducted for other breeds, particularly those with dark or black coats. In the future, we will collect a wider variety of breeds with differing coat colors to further study the performance of this model in sheep face recognition across various breeds and colors, thereby enhancing the model’s universality and robustness.

## 5. Conclusions

This study proposes a multi-modal sheep face recognition method to address the inadequate stability of sheep face recognition when relying solely on RGB single modality. This method integrates texture information from RGB data and geometric information from depth data for enhanced sheep face recognition. To achieve this, a multi-modal sheep face recognition model, termed CBAM-DRESM, was developed based on ResNet18. This model effectively combines the characteristics of the two modalities through the design of a two-stream convolutional network structure, an attention module, and multiple loss functions. Experimental results demonstrate that CBAM-DRESM outperforms the baseline network ResNet18. The enhanced CBAM-RESM network achieved an identification accuracy of 98.49%, an F1-Score of 98.39%, and an FRR of 1.50%, surpassing the performance of several backbone networks commonly utilized in recent years. In summary, the CBAM-DRESM model proposed in this paper successfully integrates the two modalities to enhance identification accuracy, exhibits strong performance in sheep face recognition, and provides a theoretical foundation and research direction for future studies in the field.

## Figures and Tables

**Figure 1 animals-15-01111-f001:**
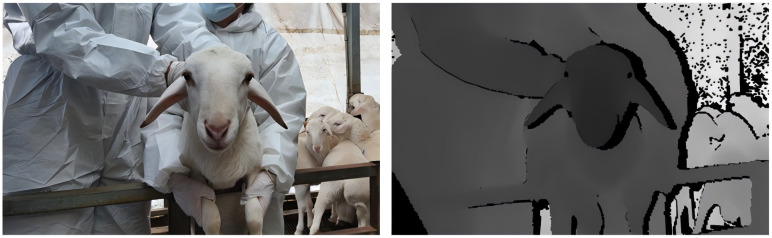
Data alignment example. RGB image (**left**) and corresponding aligned depth image (**right**).

**Figure 2 animals-15-01111-f002:**
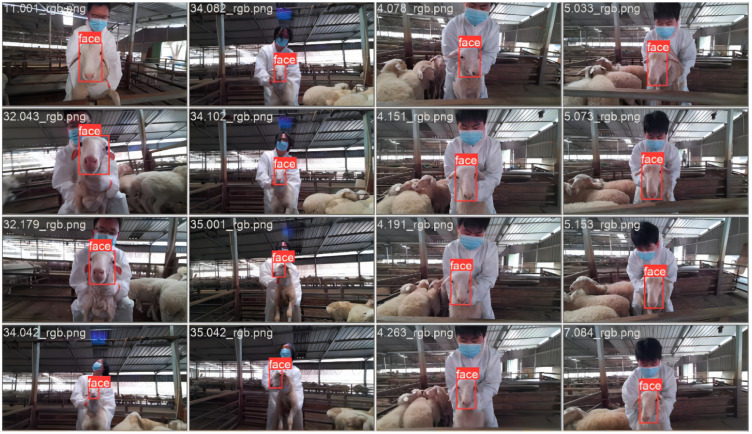
Sheep face detection example.

**Figure 3 animals-15-01111-f003:**
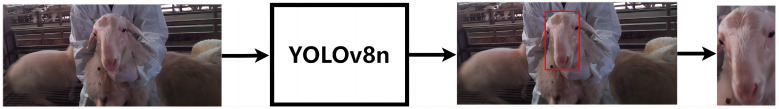
Image processing example. The red box in the figure is the final output of YOLOv8n.

**Figure 4 animals-15-01111-f004:**
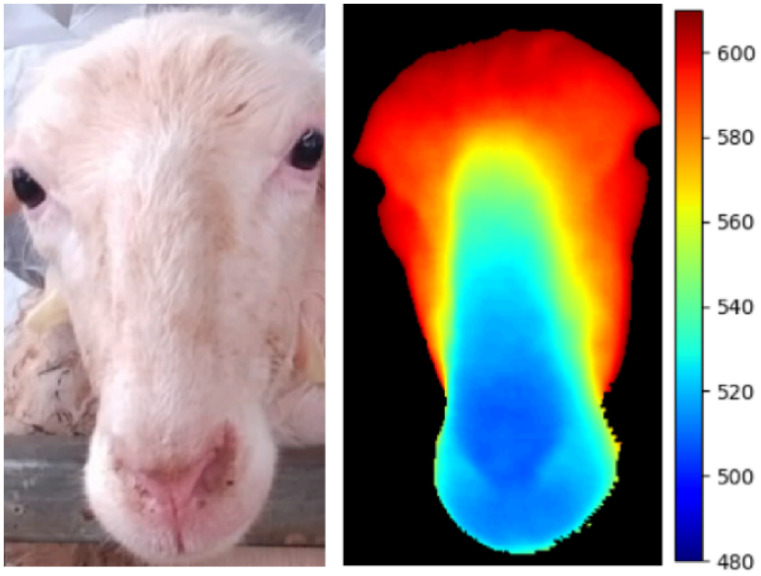
Sheep face RGB image (**left**) and corresponding depth image (**right**).

**Figure 5 animals-15-01111-f005:**
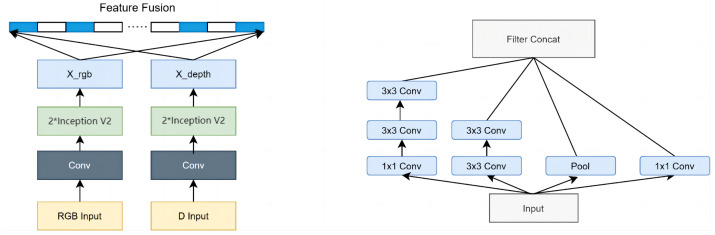
Dual branch architecture (**left**) and Inception V2 (**right**). In the left, *2*Inception V2* means two Inception V2 convolutional layers.

**Figure 6 animals-15-01111-f006:**
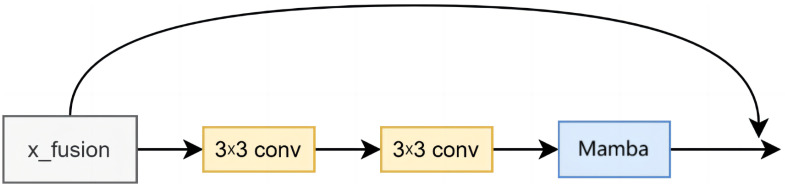
Improved Basic Block.

**Figure 7 animals-15-01111-f007:**
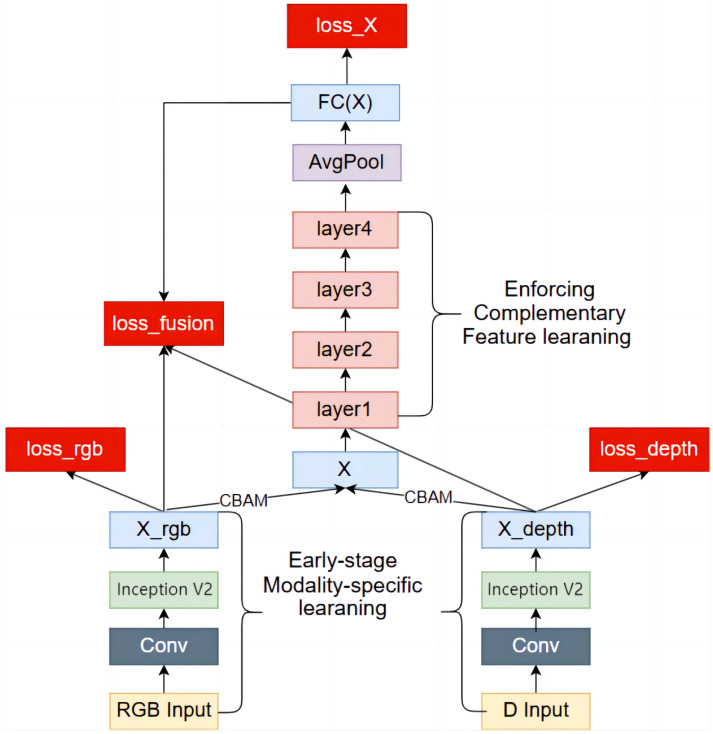
Overall structure of the CBAM-RESM model.

**Figure 8 animals-15-01111-f008:**
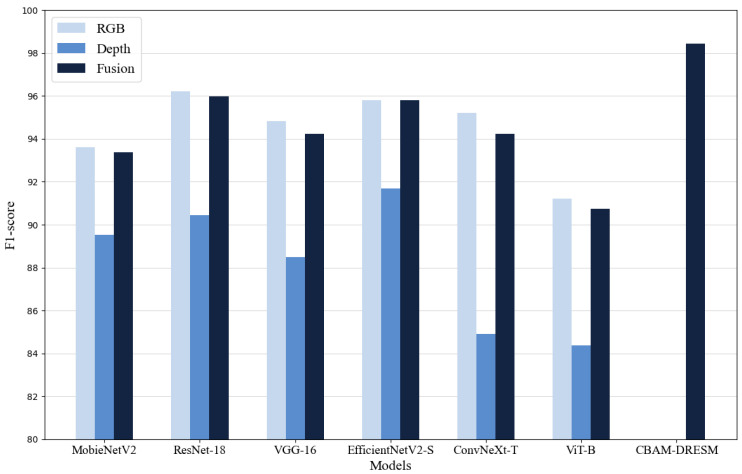
F1-Scores of the models.

**Figure 9 animals-15-01111-f009:**
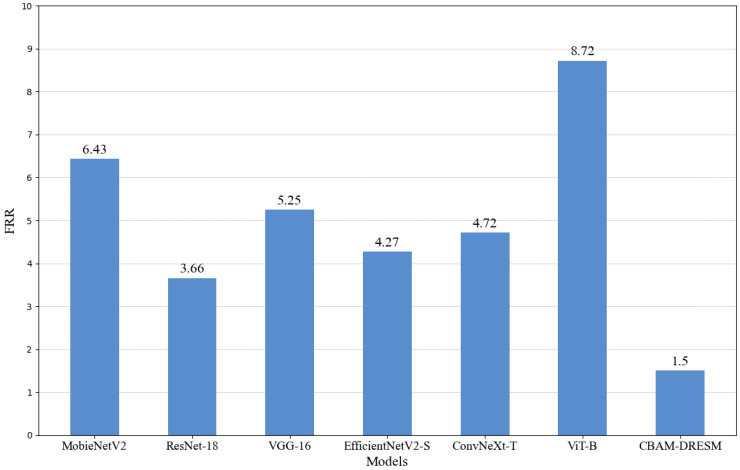
Rejection rates of different recognition models.

**Table 1 animals-15-01111-t001:** Sheep face dataset partition results.

Number of Sheep	Number of Training Sets	Number of Validation Sets
99	10,003	2501

**Table 2 animals-15-01111-t002:** Comparison of accuracy across different recognition models.

Models	RGB	Depth	Fusion
MobileNetV2	93.57%	89.55%	93.36%
ResNet18	96.32%	90.38%	96.03%
VGG-16	94.75%	88.58%	94.27%
EfficientNetV2-S	95.73%	91.69%	95.85%
ConvNeXt-T	95.20%	85.02%	94.32%
ViT-B	91.24%	84.45%	90.76%
CBAM-DRESM	-	-	98.49%

**Table 3 animals-15-01111-t003:** Ablation experiment.

Models	Identification Accuracy	F1-Score	FRR
Resnet18	96.32%	96.21%	3.66%
DualResnet18	96.75%	96.63%	3.25%
DualResnet18+CBAM	98.13%	98.01%	1.89%
DualResnet18+CBAM+Mamba	98.49%	98.39%	1.50%

**Table 4 animals-15-01111-t004:** Attention comparison experiments in the feature fusion stage.

Models	Identification Accuracy	F1-Score	FRR
DualResnet18+SE	97.12%	97.05%	2.86%
DualResnet18+ECA	97.34%	97.28%	2.65%
DualResnet18+SA	96.93%	96.92%	3.07%
DualResnet18+CA	97.88%	97.79%	2.11%
DualResnet18+CBAM	98.13%	98.01%	1.89%

**Table 5 animals-15-01111-t005:** Attention comparison experiments in the enforcing complementary feature learaning stage.

Models	Identification Accuracy	F1-Score	FRR
DualResnet18+CBAM+SE	98.05%	97.93%	1.92%
DualResnet18+CBAM+ECA	98.21%	98.10%	1.79%
DualResnet18+CBAM+SA	97.82%	97.70%	2.17%
DualResnet18+CBAM+CA	98.35%	98.24%	1.65%
DualResnet18+CBAM+Mamba	98.49%	98.39%	1.50%

## Data Availability

The datasets presented in this article are not readily available because the data are part of an ongoing study. Requests to access the datasets should be directed to the corresponding author.

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
