# Peer review of "Multi-Modality Sheep Face Recognition Based on Deep Learning"

_animals, 2025, doi:10.3390/ani15081111_

Round 1
Reviewer 1 Report
Comments and Suggestions for Authors
1. In lines 131-136, the manuscript describes that facial data was captured during data collection using a target detection algorithm. However, the resolution of the depth images differs from that of the RGB images. Please provide a detailed explanation of the methodology employed to align the depth images with the RGB images.
2. In Section 3.2, different classification models are compared. Please provide a detailed description of the versions of these models, such as MobileNet v1 or v2. Furthermore, add the classification recognition networks that have emerged in the past three years to the comparison, thus highlighting the strengths of the proposed model.
3、Please add the performance comparison of models with different attention mechanisms.
Author Response
Comments 1: In lines 131-136, the manuscript describes that facial data was captured during data collection using a target detection algorithm. However, the resolution of the depth images differs from that of the RGB images. Please provide a detailed explanation of the methodology employed to align the depth images with the RGB images.
Response 1: Thank you for pointing this out. We agree with this comment. Therefore, We have already added the data alignment method in the "Multi-modal data acquisition" subsection. This change can be found in lines 112-132.
Comments 2: In Section 3.2, different classification models are compared. Please provide a detailed description of the versions of these models, such as MobileNet v1 or v2. Furthermore, add the classification recognition networks that have emerged in the past three years to the comparison, thus highlighting the strengths of the proposed model.
Response 2: Thank you for pointing this out. We agree with this comment. Therefore, we have provided the versions and references for the discussed models. Additionally, we have incorporated another classification recognition network, the Vision Transformer (ViT)-B, which has emerged in the last three years. When compared with various recognition models, our proposed model demonstrates superior performance. These modifications can be found in lines 329-331 and Table 2.
Comments 3: Please add the performance comparison of models with different attention mechanisms.
Response 3: Thank you for pointing this out. We agree with this comment. Therefore, We have added Section 3.4, which presents comparative experiments on attention mechanisms. This section compares the performance of the CBAM attention mechanism with other attention mechanisms during the feature fusion stage, as well as the performance of the Mamba module against other attention mechanisms in the enhanced complementary feature learning stage. This change can be found in lines 368-384, Table 4 and 5.
Reviewer 2 Report
Comments and Suggestions for Authors
This study proposes a dual-branch multi-modal sheep face recognition model, combining RGB and depth data to improve recognition accuracy. The model integrates CBAM and Mamba modules for effective feature fusion and achieves high accuracy, demonstrating its potential for enhancing livestock management. Below are my comments to improve the article:
1. The Kinect V2 device has an operational range that varies approximately between 0.4 and 3 meters for depth data capture. In your study, did you conduct any evaluation to determine the optimal distance between the camera and the sheep that would allow for the best quality in RGB and depth images? If so, could you provide details on the results of this evaluation? If not, do you consider this aspect to be relevant for future work, especially when using more modern devices with different operational ranges?
2. In the study, it is observed that the images of the sheep presented are predominantly white. However, in practice, there are sheep of other colors, such as black or darker shades. Given that RGB images can be affected by variability in coat color, how does the proposed model ensure consistent performance on sheep with darker or non-light colors? I understand that the use of depth images helps mitigate this issue, but could you provide more details on how the model handles this variability and whether specific tests were conducted with sheep of different colors?
3. The Kinect is a device that has been widely used in various applications, but it has been discontinued, and there are now new cameras of this style available. Is there any proposal for a new camera, and what would be the challenges for its use? I would like this to be mentioned in the article since it is not currently addressed.
4. The Discussion Section can be further expanded, more comparisons about the conclusions in this paper and in relevant research can be added to find their similarities and differences.
Author Response
Comments 1: The Kinect V2 device has an operational range that varies approximately between 0.4 and 3 meters for depth data capture. In your study, did you conduct any evaluation to determine the optimal distance between the camera and the sheep that would allow for the best quality in RGB and depth images? If so, could you provide details on the results of this evaluation? If not, do you consider this aspect to be relevant for future work, especially when using more modern devices with different operational ranges?
Response 1: Thank you for pointing this out. We have specifically addressed and provided insights on this issue in the discussion section. Prior to the experiment, we observed that the depth maps of human faces captured by the Kinect V2 camera were more complete and detailed when the target was positioned within a distance range of 0.8 to 2 meters from the camera. Consequently, we adopted this distance range as the shooting distance for subsequent research. However, this conclusion has yet to be systematically evaluated. In future work, we will conduct more comprehensive experiments to thoroughly investigate the optimal shooting distance for 3D cameras in sheep face recognition, aiming to provide more precise guidance for practical applications.
Comments 2: In the study, it is observed that the images of the sheep presented are predominantly white. However, in practice, there are sheep of other colors, such as black or darker shades. Given that RGB images can be affected by variability in coat color, how does the proposed model ensure consistent performance on sheep with darker or non-light colors? I understand that the use of depth images helps mitigate this issue, but could you provide more details on how the model handles this variability and whether specific tests were conducted with sheep of different colors?
Response 2: Thank you for pointing this out. The sheep breeds primarily collected in this study are Hu sheep, Junken White sheep, and White Suffolk sheep, all of which predominantly have white coats. Systematic testing has not yet been conducted for other breeds, particularly those with dark or black coats. In the future, we will collect a wider variety of breeds with differing coat colors to further study the performance of this model in sheep face recognition across various breeds and colors, thereby enhancing the model's universality and robustness.
Comments 3: The Kinect is a device that has been widely used in various applications, but it has been discontinued, and there are now new cameras of this style available. Is there any proposal for a new camera, and what would be the challenges for its use? I would like this to be mentioned in the article since it is not currently addressed.
Response 3: Thank you for pointing this out. Given that the Kinect V2 has been discontinued, our team has tested various types of 3D cameras to ensure the sustainability of the research. The cameras evaluated include Hikvision MV-DT01SDU, Intel RealSense D435F, and Orbbec Femto Bolt. After comprehensive consideration of factors such as performance and cost-effectiveness, we ultimately selected Orbbec Femto Bolt as a replacement for the Kinect V2. This camera not only offers excellent performance but is also compatible with the underlying architecture of the Azure Kinect SDK, effectively reducing the technical barriers for system migration and providing strong support for the smooth progression of the research.
Comments 4: The Discussion Section can be further expanded, more comparisons about the conclusions in this paper and in relevant research can be added to find their similarities and differences.
Response 4: Thank you for pointing this out. In accordance with your suggestion, we have added and expanded the discussion section.This change can be found in lines 386-396.